# Testing of Polymer Composites for Manufacturing of Sprayer Nozzles

**DOI:** 10.3390/polym16040496

**Published:** 2024-02-10

**Authors:** Victoria E. Slavkina, Maksim A. Mirzaev, Anton M. Kuzmin, Alexey I. Kutyrev, Sergey P. Tuzhilin, Vyacheslav A. Denisov, Yuriy V. Kataev

**Affiliations:** 1Department of Agricultural, Federal State Budgetary Scientific Institution “Federal Scientific Agroengineering Center VIM” (FSAC VIM), 109428 Moscow, Russia; mirza.pochta@gmail.com (M.A.M.); alexeykutyrev@gmail.com (A.I.K.); sptuzh@mail.ru (S.P.T.); va.denisov@mail.ru (V.A.D.); ykataev@mail.ru (Y.V.K.); 2Department of Ogarev, Mordovia State University, 430005 Saransk, Russia; kuzmin.a.m@yandex.ru

**Keywords:** nozzle, agricultural sprayer, polyoxymethylene, disperse fillers, chemical resistance, hydroabrasive wear, viscosity

## Abstract

Wear is the leading cause of nozzle failure. The durability of the nozzle is affected by the material it is made from. Traditional materials are ceramics, stainless steel, brass, and polymers. One of the possible ways to improve the wear resistance of polymer nozzles is through the incorporation of dispersed fillers into them. This paper presents the results of testing polymer composites for their chemical resistance to pesticides, examining the effects of different types and amounts of fillers on the chemical and abrasion resistance. When silicon carbide was used as a filler, the strength increased by 30.2%. The experiments on chemical resistance to pesticides revealed that the nature, shape, and volume content of filler particles do not significantly affect the resistance of the compounds obtained. Tests on hydro-abrasive wear have shown that graphite and silicon carbide are effective fillers capable of reducing wear by up to 7.5 times. Based on previous research, it is recommended to use a composite compound with 15% volume of silicon carbide for nozzle manufacturing.

## 1. Introduction

The Food and Agriculture Organization of the United Nations (FAO) reports that plant pests and diseases cause global losses of approximately 20–25% of potential food crop yields each year. In Russia, up to 25% of agricultural products are lost, meaning that every fifth hectare fails to yield its potential [1,2].

Chemical plant protection is one of the main factors that can address this problem. The agroecosystem currently heavily relies on pesticides to meet the increasing demand for food. When used at the right dosage, pesticides can increase yields and improve the quality of field produce. Without pesticides, fruit crop yields would decrease by 78%, vegetable crop yields by 54%, and cereal yields by 32%. Pesticide treatment of crops should be carried out at least twice from plant development to maturity before harvesting [3,4,5,6,7,8,9,10].

Currently, the most common method of chemically treating agricultural crops is through spraying. The efficiency of spraying is determined by the timing of application, degree of coverage, and dosage of the working solution. Pesticides are applied to the crops in the form of a working solution. The nozzle is a crucial element that affects the quality of spraying [11,12,13].

A nozzle is a device that splits the working fluid into drops of a certain size, ensuring the formation of symmetrical atomizing plumes. Currently, the Russian market offers nozzles from various manufacturers, including “Lechler” (Metzingen, Germany), “Teejet” (Wheaton, IL, USA), “Hypro” (Minneapolis, MN, USA), “Combo-Jet” (Lexington, KY, USA), “AgroPlast” (Sawin, Poland), “Albuz” (Evreux, France), “Nozal” (Noisiel, France), “Hardi” (Norre Alslev, Denmark), “ARAG” (Reggio nell’Emilia, Italy), and others.

Nozzle failure is often caused by wear. The abrasive effects of sand, kaolin, and other impurities in the water, as well as abrasive pesticides (such as insecticides made from natural materials like pumice, corundum, or diatomite), is the cause of wear on the working solution [14,15,16]. The change in nozzle tip shape and the effect on spray angle is shown in Figure 1.

When selecting a nozzle, it is crucial to consider its wear resistance. This refers to the amount of time the nozzle can operate without significant changes in output, torch angle, or liquid distribution quality. The wear resistance is directly dependent on the material of the nozzle. Ceramics, stainless steel, brass, and polymers are currently used as materials [17,18,19,20].

Polymer nozzles are a cost-effective alternative to ceramic, steel, or brass nozzles. Modern polymers are technologically advanced and as durable as stainless steel. While brass or bronze nozzles are inexpensive, they wear out relatively quickly. According to the research conducted by SGS UK LTD Saint-Globain AC (Courbevoie, France), at a pressure of 0.275 MPa and a water content of 2.5% kaolin, a slotted nozzle made of brass reaches a flow rate 10% higher than that given in the table after 10 h; stainless steel after 30 h; and polyvinylidene fluoride after 40 h. After 50 h, the flow rate of the polyacetal nozzles increases by 8%, and that of the ceramic nozzle by 5% [21,22,23].

Figure 2 displays the comparative wear resistance of materials for nozzles.

Farmers are transitioning from ceramic to polymer nozzles due to their lower cost and reduced maintenance requirements. The increasing use of complex mixtures of chemicals and microfertilisers, as well as the filling of nozzles from water bodies where the water does not meet the necessary requirements, leads to the need for more frequent cleaning of nozzles. Cleaning ceramic nozzles with a wire or awl can cause the edges to break, requiring the entire set to be replaced. Ceramic nozzles are more expensive than polymer ones, with one set costing as much as 3–4 polymer sets. Additionally, ceramic nozzles have low frost resistance.

However, not all polymers are suitable for producing agricultural nozzles. The polymer must have exceptional chemical and abrasion resistance and be specifically adapted for high-precision moulding [24]. The most commonly used polymer for nozzle production is polyoxymethylene, a crystalline polymer material with a degree of crystallinity exceeding 70%. Because of the strong -C-O- structure in the main chain, POM maintains stable chemical composition and excellent mechanical properties, including high strength and rigidity, favourable impact/slip resistance, and durability.

In reference [25], it was noted that injection moulding technology can be used to mould gears from polyacetal. Reference [26] demonstrated that Hostaform polyacetal is suitable for manufacturing screw gears, conveyor rollers, holders, sliding bearings, and parts of conveyor chains. Work [27] reported an increase in the durability of the bearing unit of the support roller of a potato digger by replacing a rolling bearing with a sliding bearing made from the polymeric material polyacetal. Polyacetal has better wear resistance than stainless steel by a factor of 2, and than brass parts by over 30 times, but is inferior to ceramics in this regard [28].

Dispersed fillers can be used to solve this problem [29]. Introducing them into polymer composites can alter their mechanical, physical, and other properties, allowing the final material to have the necessary characteristics [30,31,32,33].

To enhance the wear resistance of nozzles, it is necessary to increase the hardness of the material they are made of. This can be achieved by introducing fillers into the polymer that have a higher hardness than the polymer itself. Examples of such fillers include glass microspheres, ceramic particles, and solid fibres [34].

Glass microspheres are commonly used to fill various types of polymers. These microspheres have a high hardness of 5.5 on the Mohs scale. However, this comes at the cost of reduced plasticity and impact toughness, which is a common trade-off when introducing dispersed fillers [35,36,37].

Ceramic particles, including oxides, borides, and carbides of various substances, are known for their high hardness. Reference [38] showed that incorporating these particles into polymers can significantly improve wear resistance. Among these particles, nanoscale silicon carbide has been found to have the most significant effect on wear resistance.

Different types of fibres are commonly used as reinforcing fillers in composite material manufacturing. Fibres possess several desirable properties, including high hardness; fungal, chemical, and heat resistance; exceptionally high tensile strength; ideal elasticity up to fracture; high specific surface area; and the presence of hydroxyl groups. These properties allow for complete wetting of the filler by polymer binders and high strength of adhesion of the matrix with the fibre at the interface [39,40,41]. The fibre filler content in thermoplastics typically ranges from 15% to 40%.

Polymer composite materials are utilized to manufacture parts that operate under friction conditions. These materials contain special fillers that can reduce the coefficient of friction. Typically, these fillers have a layered structure, with graphite and molybdenum disulphide being the most common. The antifriction fillers constitute 2–10% of the mass of the polymer composite material and have a minimal impact on other properties of the polymer [42].

Graphite is a mineral and one of the allotropic modifications of carbon. This material has high self-lubricating properties because it adheres well to various surfaces and allows particles to slide easily [43].

Molybdenum disulphide is an effective antifriction additive due to its layered structure and low coefficient of friction. However, adding it as a filler significantly increases the final cost of the product [44,45].

Taking into account the aforementioned points, the aim of the presented research paper was to find the optimal composition of composite material based on polyoxymethylene and to analyse the mechanical and tribological properties of composites containing dispersed modifiers such as glass fibre, graphite, and silicon carbide.

The scientific novelty of this paper lies in the regularities of the technological processes involved in mixing and compounding the components of thermoplastic composites. Additionally, the paper presents the results of mechanical tests, hydro-abrasive wear tests, and tests on the chemical resistance of composites to herbicides, fungicides, and insecticides.

The use of modifiers such as glass fibre, graphite, and silicon carbide in polyoxymethylene has been attributed to the poor performance of previously used materials for spray nozzles. This limits their scope of application.

## 2. Materials and Methods

### 2.1. Compositions and Manufacture of Test Samples

This paper analyses polymer composites for agricultural nozzles, focusing on their chemical resistance to agrochemicals, water abrasion, mechanical strength, and toughness.

The polymer matrix chosen for this work was polyoxymethylene MASCON POM 27 (Rusplast Ltd., Noginsk, Russia). Dispersed fillers such as glass fibre (Energon, Saint-Petersburg, Russia), graphite (GraphiteService, Chelyabinsk, Russia), and silicon carbide (JUK F600, 63-64C, Moscow, Russia) were selected based on an earlier analytical review.

The investigated samples and their compositions are listed in Table 1. The equilibrium average of volume filling was calculated based on the recommended values for all types of fillers used.

The polyacetal granules were dried in an oven at 100 °C for 6 h before being placed into an intermittent rotary mixer for specimen manufacturing. The mixing process was carried out at a temperature of 170 °C and rotor speed of 50 rpm for 40 min. The resulting mixture was then crushed to obtain granules (Figure 3).

To obtain test samples, the granulate was pressed into 2 mm thick plates using a hydraulic press at a temperature of 180 °C for 30 min. The plates were then cooled under the press for an hour to ensure uniform cooling throughout the sample. Finally, the desired geometry of the test samples was cut out using a laser cutting machine.

### 2.2. Parameters for Mechanical Strength Test

The composite samples’ strength was tested using a tensile testing machine, I1147M-50-01-1 (Tochpribor-KB LLC, Ivanovo, Russia) [46].

The test involved stretching the specimen along its longitudinal axis at a constant velocity of 5 mm/min. During this process, the load withstood by the specimen and its elongation were measured. The dimensions of the samples are listed in Table 2.

The experiments’ results were accompanied by confidence intervals (Equation (1)), assuming a 95% confidence level in the calculations.
(1)$$\bar{X}\;-\;t_{\alpha ,n-1}\;\times \;\frac{\hat{\sigma }}{\sqrt{n}}\;<\;\mu \;<\;\bar{X}+t_{\alpha ,n-1}\;\times \;\frac{\hat{\sigma }}{\sqrt{n}}$$
where $\bar{X}$—arithmetic mean; $t_{\alpha ,n-1}$—Student’s *t*-test with distribution α; $\hat{\sigma }$—standard deviation.

### 2.3. Test of Resistance to Chemicals

The resistance of composites to aggressive chemical agents was tested following standard recommendations. Sixty square samples, each with a side length of 60.0 mm and a thickness of 1.0 mm, were used for the experiment.

Solutions of the herbicide “Lazurit”, the fungicide “Rakurs”, and the insecticide “Borei-Neo” were used as chemical agents. Tests were conducted using solutions of standard concentration recommended for crop treatment, as well as solutions with concentrations of chemicals five times higher to expedite the testing process. The calculation values used are shown in Table 3.

The resistance of the specimens to the selected agents was evaluated by measuring the change in mass and hardness during soaking in the studied solutions. Mass was measured using analytical scales (A&D Co. LTD., Tokyo, Japan) with a discreteness of 0.1 mg, while hardness changes were determined using a Shore D hardness tester (Vostok-7, Moscow, Russia) with an error of ±2. Hardness was measured under a 5 kg load for 10 s. Measurements were taken at three points on the specimen, and then the average of the obtained values was calculated.

The hardness and weight values of each sample were measured prior to testing. Each sample was weighed three times, and the data were recorded in a final table from which the arithmetic mean was calculated. The samples were then fully immersed in the test liquid for a specified period of time. Daily mass and hardness measurements were taken during the first week. Measurements were taken weekly during the 15-week test period, each time a new solution was prepared for the immersion process.

### 2.4. Hydro-Abrasion Experiment

An analysis of machines used for testing water-abrasive wear resistance has shown that it is feasible to mount test samples on a rotor and immerse them in an abrasive liquid [47,48,49]. This causes the abrasive particles to move and wear down the test samples. The samples were fully submerged in the liquid in a beaker and the rotor was rotated at a speed of 100 rpm (Figure 4).

The specimens tested were 1 × 1 cm in size. To accelerate the wear process, the research was conducted with the addition of 30% sand. The tests were carried out for unfilled POM samples and samples containing 15% of each type of filler. Hydro-abrasive wear was evaluated by the change in the mass of the samples before and after the test. An A&D GR-200 analytical scale (A&D Co. LTD., Tokyo, Japan) was used to measure the mass. As polymers are hygroscopic, after the test, the samples were placed in an oven at 100 °C for 2 h to remove water from them.

### 2.5. Optical Microscopy

In order to obtain a complete picture of the nature of the interaction between the polymer matrix and the fillers, an optical microscope, OLYMPUS BX53M (Olympus Corporation, Tokyo, Japan), was used. The surfaces of the samples were examined after hydroabrasive wear. The studies were carried out at 4- and 20-times magnification.

### 2.6. Planned Three-Factor Experiment to Determine the Viscosity of the Composite

Since the nozzles of agricultural sprayers are manufactured by injection moulding, one of the most important indicators of the polymer composites studied is viscosity. The study of the rheological properties of the polymer composites was carried out using the extrusion plastometer PTR-LAB-02 (“Laboratory equipment and devices”; Saint-Petersburg, Russia). The principle of the plastometer is to force the polymer through a capillary of a certain diameter by means of a piston under a certain load (Figure 5). The diameter of the capillary is 2 mm. To measure the viscosity index, the temperature and load values were varied. The time taken for the polymer to pass through the capillary was recorded during the test.

The temperatures of the samples varied from 165 °C to 195 °C, with a step variation of 15 °C. Weights of 1.2; 2.6; and 3.8 kg were used. The tests were carried out on polyacetal samples filled with 10, 15, and 20% silicon carbide. After the experimental investigation, the effective viscosity was calculated. To accomplish this, we first calculated the melt pressure for each of the weights used according to Equation (2):P = F/S_k_, Pa(2)
where F—load from the piston; S_k_—cross-sectional area of the chamber.

The load on the piston was calculated using Equation (3):F = m·g, N(3)
where m = 1.2; 2.6; 3.8 kg—the weight of the load used.

The cross-sectional area of the chamber was calculated using Equation (4):(4)$$S_{k}=\frac{\pi D_{\mathrm{cam}}^{2}}{4},m^{2}$$where D_cam_—chamber diameter.

The melt volume flow q and the linear velocity of the rod ν for each measurement were calculated using Equations (5) and (6):q = ν·S_k,_ m^3^/c(5)
(6)$$v=\frac{h}{t}\;m/c$$
where h—distance between the marks on the piston rod; t—time for the piston rod to move from the bottom mark to the top mark.

Shear stress τ and average shear rate γ in the capillary were calculated using Equations (7) and (8):(7)$$\tau \;=\frac{P\cdot r}{2\cdot L},\;\mathrm{Pa}$$
(8)$$\gamma =\frac{q}{\pi r^{3}},\;C^{-1}$$where L—capillary length; r—capillary radius.

The viscosity value was calculated according to Equation (9):(9)$$\eta =\frac{\tau }{\gamma }\;Pa\cdot c,$$

The data processing programme of the PFE 2^3^ type three-factor design experiment was used. To compensate for the influence of random errors, n = 3 parallel experiments were performed. In order to improve the accuracy of the result and to bring the calculations into a uniform form, the experiment was conducted with uniformity of experiments, i.e., all rows of the design matrix had the same number of parallel experiments [50,51,52].

## 3. Results and Discussion

### 3.1. Results of the Mechanical Strength Tests

In order to study the effect of reinforcing the polyacetal with dispersed fillers on its properties, its strength properties were tested. The results are shown in Figure 6.

As can be clearly seen from the data obtained, the addition of dispersed fillers led to an increase in tensile strength. The highest increase in the strength index was achieved at 15% filler by volume for all types of filler. Dispersed fillers such as glass fibres have high strength and stiffness compared to polymers. When added to the polymer matrix, they strengthened the material and improve its mechanical properties. Dispersed fillers helped to distribute the mechanical load evenly throughout the material, preventing stresses from building up in specific areas and increasing the overall strength of the composite. This effect was most pronounced in composites containing silicon carbide particles as fillers. When silicon carbide was used as a filler, the strength increased by 30.2%. When glass fibre and graphite were added as fillers, the tensile strength increased by 26.8 and 27.4%, respectively.

### 3.2. Chemical Resistance Test Results

The results of the chemical resistance tests were processed using Google Colaboratory service. Figure 7 shows plots of the hardnesses of pure POM samples as a function of the type and concentration of pesticide used.

By analysing the results obtained, it can be concluded that increasing the concentration of the solution by a factor of 5 relative to the standard had no significant effect on the rate of change of hardness of the materials tested. A similar pattern was observed for the change in mass.

The results of the hardness of the samples when soaked in crop protection chemicals, depending on the type of filler, are shown in Figure 8. The samples used for analysis were those immersed in solutions of higher concentrations.

As can be seen from the data obtained, neither pure POM nor compositions based on it underwent significant changes in hardness when aged in solutions of chemical pesticides. On average, the change in hardness varied between 0.5 and 2.5%.

The results for the weights of the samples while soaked in crop protection products are shown in Figure 9.

The average weight gain of all samples after 15 weeks in the solutions was approximately 0.1 to 4.5% of the mass of the test sample. The increase in the mass of the samples is attributed to the residual liquid retained within the samples even after drying, as polyacetal is a hygroscopic material.

### 3.3. Hydroabrasion Test Results

Based on the results of the mechanical properties and chemical resistance tests presented earlier, it was found that the best properties were obtained from blends containing 15% filler. Therefore, these compositions were used for the hydroabrasive fracture tests. As a result of the tests, the mass of the samples was measured after 1 and 5 h of testing. The results are shown in Table 4.

Based on the results obtained, the change in the mass of the samples as a result of the test was calculated, and an average value was obtained for the series of two samples. The volumetric wear was then calculated, taking into account the difference in density between the tested samples. The results are presented as a histogram in Figure 10.

As can be seen from the results obtained, the polyoxymethylene-based composite samples filled with glass fibres showed the maximum wear. This result is likely due to the fact that the sand particles tore the glass fibres out of the composite during rotation and, as the fibres were of considerable length, the volume of material removed as a result of this effect was also large. After 1 h of testing, the wear rate of the composites containing graphite and silicon carbide was three times lower than that of pure POM. After 5 h of testing, the wear pattern had changed—the pure POM samples showed an increase in wear by a factor of almost 5, while the graphite samples increased by a factor of only 2, and the silicon carbide composites by a factor of 13.

For example, when comparing the wear of pure POM and composite samples after 5 h of testing, it was found that silicon carbide filler reduced the wear by 1.8 times and graphite by 7.5 times. The reduction in wear of the silicon carbide samples can be explained by the high hardness of the filler used, which acted as a protection against scratching of the material by sand particles. In the case of graphite, however, the mechanism of protection against hydroabrasive wear cannot be the same, because graphite is a soft filler. Graphite particles can act as a hard lubricant, allowing sand particles to slide over the surface of the material without shearing it.

According to the results of nozzle wear tests, graphite and silicon carbide are effective fillers that can reduce wear by up to 7.5 times. However, previous data on abrasive wear of composite blends, presented in [53], showed that the most wear-resistant blend was based on polyacetal filled with 15% silicon carbide. Therefore, further studies were carried out with a compound based on polyacetal filled with silicon carbide.

### 3.4. Results of Microscopy

Figure 11 shows photographs taken by means of optical microscopy at 4× (left photo) and 20× (right photo) magnification.

The optical microscopy results show that, in the samples with polyacetal filler, there was orientational stretching of the filler particles, mainly at the corners. In the graphite-filled samples, the filler was distributed between the matrix macromolecules and the filler particles were able to aggregate, apparently due to electrostatic forces. It appears that a so-called “lattice within a lattice” can be formed, which would contribute to the improvement of the strength properties due to additional bonds and the formation of an additional “framework” element. The samples filled with silicon carbide showed a chaotic distribution of filler particles between the matrix macromolecules, with the particles coming to the surface of the material. This can lead to abrasion of the aggressive factor acting on the material during the test, which explains the superior resistance of these samples to hydroabrasive wear.

### 3.5. Test Results of a Full-Factor Planned Experiment

In order to study the influence of factors and their interactions (temperature, load, and filler content) on the viscosity index of a composite based on polyoxymethylene filled with silicon carbide, a complete PFE 2^3^ factor experiment was carried out. The parameters set and the viscosity results obtained are shown in Table 5.

General form of regression equation:$$y=b_{0}+b_{1}x_{1}+b_{2}x_{2}+b_{3}x_{3}+b_{12}x_{1}x_{2}+b_{13}x_{1}x_{3}+b_{23}x_{2}x_{3}+b_{123}x_{1}x_{2}x_{3}$$

The calculated coefficients allow the regression equation (factor model) to be written in general terms:

y = 1154.67 + (−5753.9988) × x_1_ + 457.745 × x_2_ + 20.5849 × x_3_ + (−234.055) × x_1_ × x_2_ + (−1.5999) × x_1_ × x_3_ + 13.7199 × x_2_ × x_3_ + 61.2800 × x_1_ × x_2_ × x_3_

A statistical evaluation of the planned experiment was then performed with a confidence level of 0.95 (Table 6).

Parameter values in the table: Variance is a measure of the spread of all the data in the sample; the Cochran test is used to test for homogeneity of variance between groups; homogeneity of variance is important for the reliability of analysis of variance and indicates the similarity of variance between groups or samples; adequacy variance assesses how well the model fits the data; reproducibility variance measures how well the results of a study can be reproduced; and the Fisher test is used to determine the statistical significance of the results of a study.

In this experiment, the calculated value of criterion *G_p_* did not exceed the table value of criterion *G_T_* at a significance level of 5%, being 0.5157 under the conditions of conducted experiments:0.8294 > 0.5157, i.e., *G_P_* > *G_T_* => dispersions are heterogeneous.

After testing the statistical significance of the calculated factor model coefficients using Student’s *t*-test, tp was calculated for each factor and compared with the tabulated value. A factor is significant if tp > tT for the accepted significance level of 5%, indicating its importance in the model. The final equation of the simplified factorial mathematical model allowed us to formulate specific conclusions about the properties of the investigated composites.
y = −2491.483 + 15.3659 × x_1_ + 4209.605 × x_2_ + 282.831 × x_3_ + (−21.4305) × x_1_ × x_2_ + (−1.5713) × x_1_ × x_3_ + (−113.1324) × x_2_ × x_3_ + 0.6285 × x_1_ × x_2_ × x_3_

We used Fisher’s criterion to estimate the variances. Heterogeneity of variance is important in regression analysis and may affect the reliability of the results, requiring additional analysis and possible adjustment of the model. Since Fp < 1, the mathematical description of the response function by the regression equation is considered adequate.

Further processing was carried out using Google Colaboratory service to determine minimum viscosity values at minimum temperatures and loads for a polyacetal-based compound filled with 15% silicon carbide (Figure 12).

The extremum of the response function lies within the range of variation of the factor variables. The extreme value is y_opt_ = 360.539. An extremum of the response function corresponds to the values of the factors: x_1_ = 194.88 and x_2_ = 1.2.

## 4. Conclusions

In this study, the possibilities of using polymer composites based on polyacetal with dispersed fillers for the manufacture of agricultural sprayer nozzles were considered. Tensile strength tests have shown that the addition of dispersed fillers leads to an increase in tensile strength. When silicon carbide was used as a filler, the strength increased by 30.2%.

The test results also showed that the composite samples exhibited the same resistance to aggressive chemical environments as pure POM used in the manufacturing of agricultural nozzles. It can, therefore, be concluded that the composite materials considered are acceptable for the manufacturing of nozzles in terms of chemical resistance. It has been shown that the use of dispersed fillers such as graphite and silicon carbide can reduce the hydroabrasive wear of the material by up to 7.5 times.

To test the influence of factors and their interaction processes on the viscosity index, a three-factor design PFE 23 was performed. The design was statistically evaluated at the 95% confidence level. The recommended parameters for material processing were determined: 195 °C at a load of 1.2 N.

It is, therefore, recommended to use a polyacetal-based polymer composite with 15% silicon carbide as a disperse filler for the manufacturing of agricultural nozzles.

## Figures and Tables

**Figure 1 polymers-16-00496-f001:**
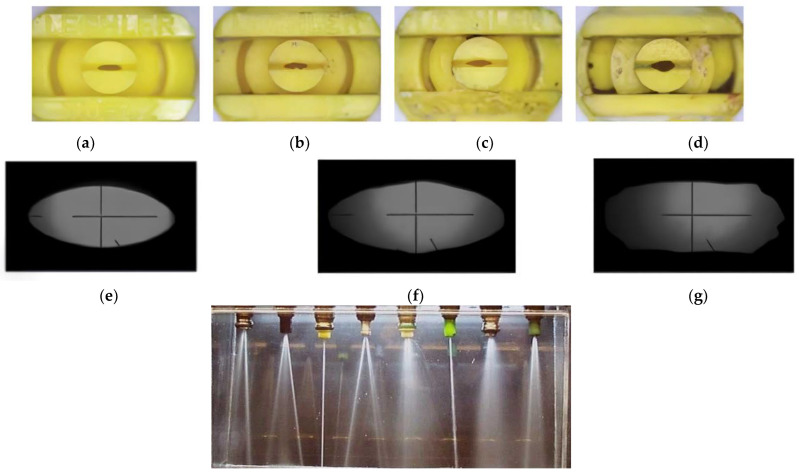
The spray pattern of an agricultural nozzle can change due to wear: (**a**) new nozzle; (**b**) after 40 h of use; (**c**) after 70 h of use; (**d**) after 180 h of use; (**e**) original shape; (**f**) partially worn nozzle; (**g**) catastrophic wear.

**Figure 2 polymers-16-00496-f002:**
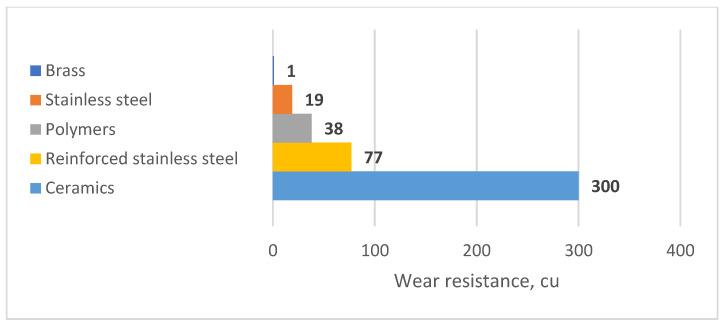
Relative wear resistance of different nozzle materials.

**Figure 3 polymers-16-00496-f003:**
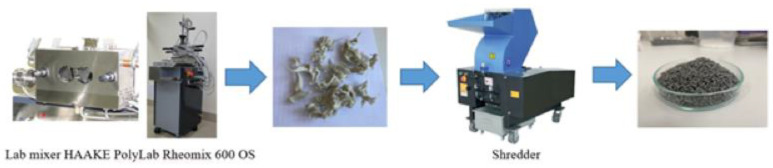
Manufacturing of composite granules.

**Figure 4 polymers-16-00496-f004:**
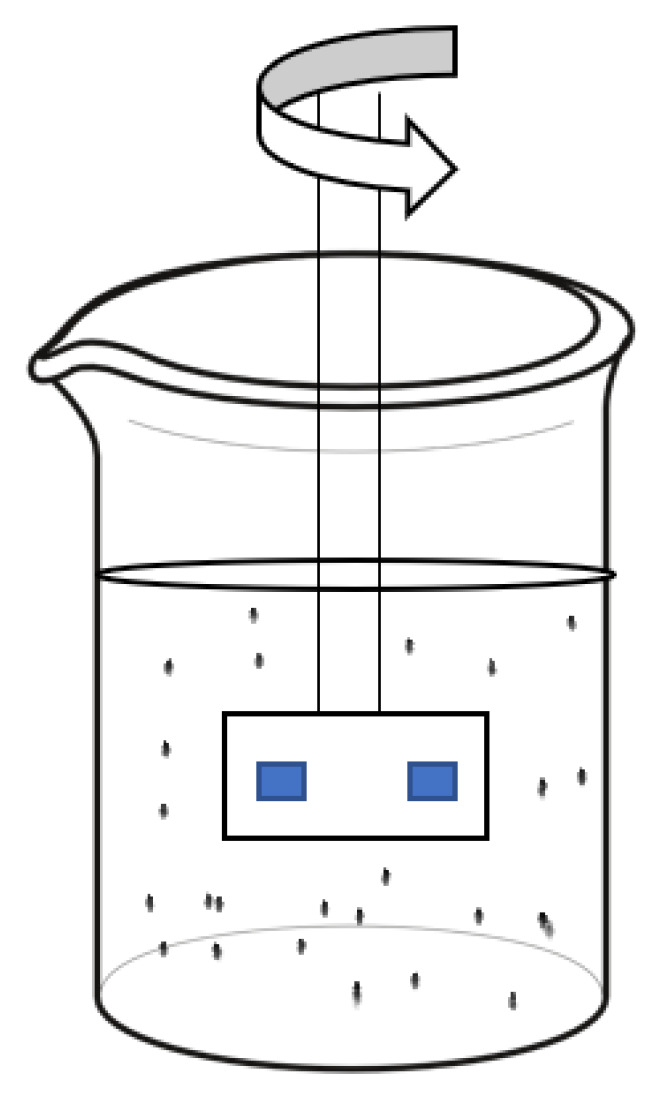
Hydro-abrasive wear testing scheme.

**Figure 5 polymers-16-00496-f005:**
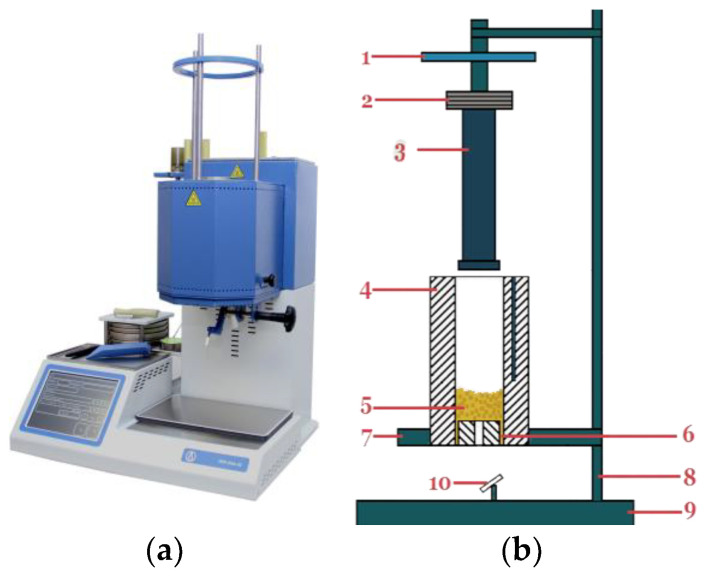
PTR-LAB-02 installation: (**a**) optical image of the device; (**b**) diagram of the device: **1**—hand wheel; **2—**set of weights; **3**—piston; **4**—heating chamber; **5**—material to be tested; **6**—capillary; **7**—heating chamber base plate; **8**—column; **9**—device base plate; **10**—viewing mirror.

**Figure 6 polymers-16-00496-f006:**
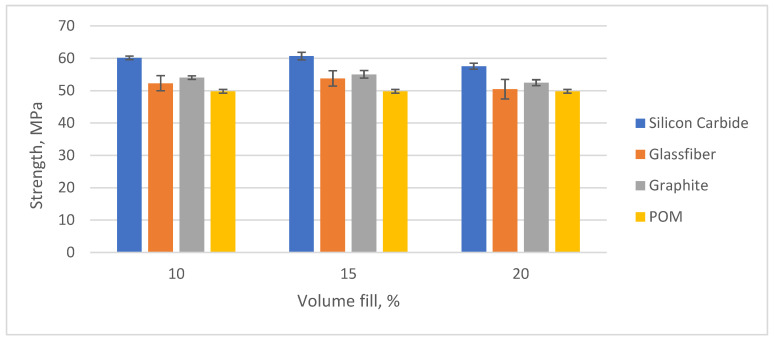
Mechanical strength value.

**Figure 7 polymers-16-00496-f007:**
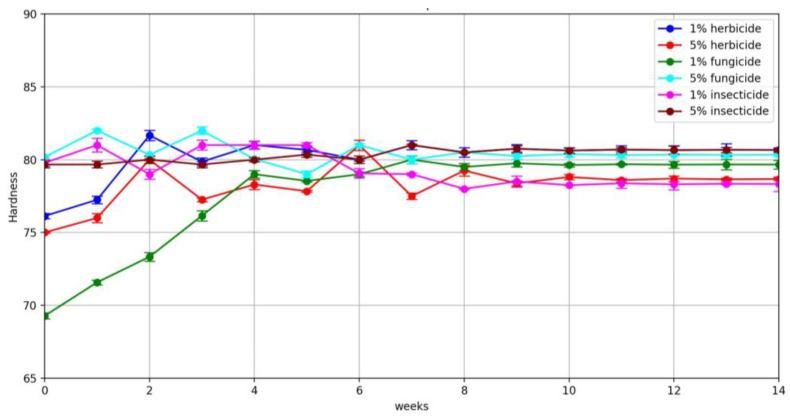
Change in hardness (Shore units) of pure POM samples depending on type, concentration, and dwell time.

**Figure 8 polymers-16-00496-f008:**
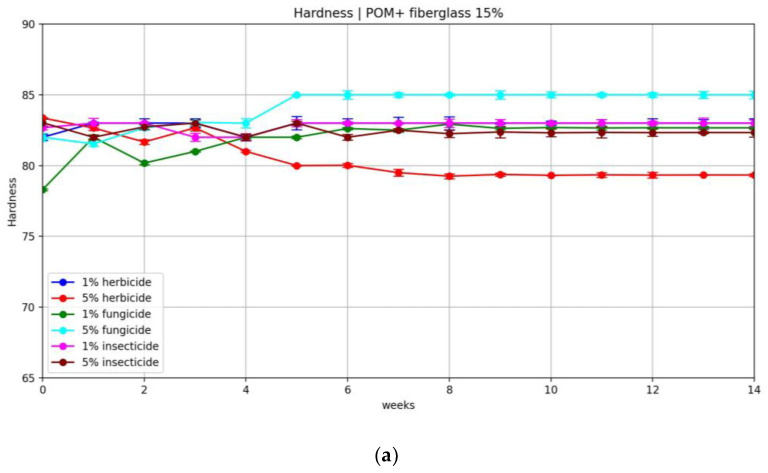
Changes in hardness (Shore units) of pure POM and POM-based samples as a function of soaking time: (**a**) herbicide solution; (**b**) fungicide solution; (**c**) insecticide solution.

**Figure 9 polymers-16-00496-f009:**
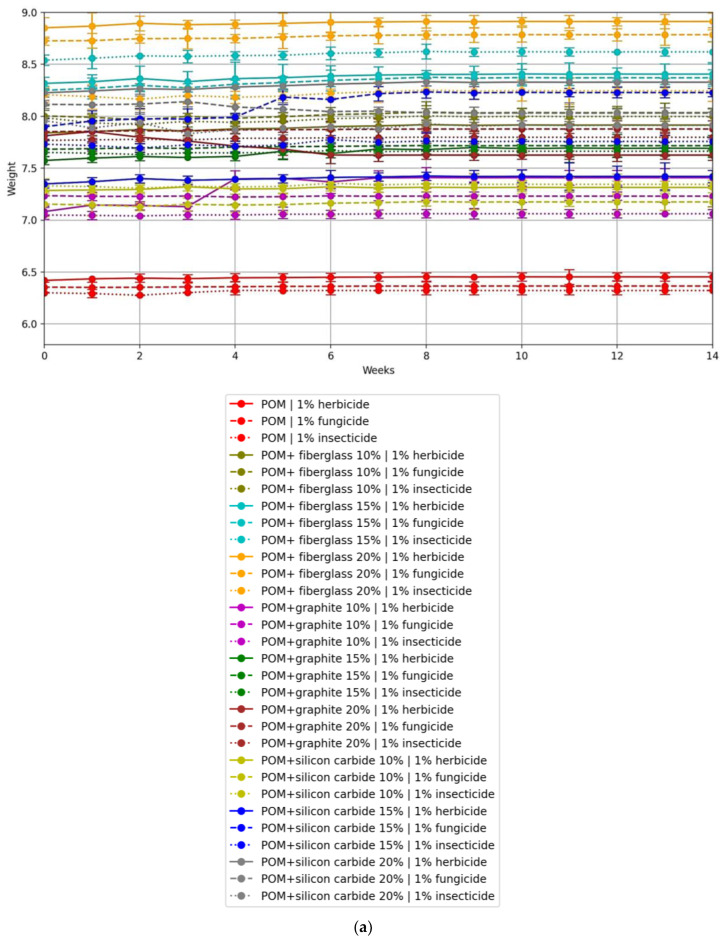
Change in mass (grams) of samples (**a**) in solutions of 1% concentration; (**b**) in solutions of 5% concentration.

**Figure 10 polymers-16-00496-f010:**
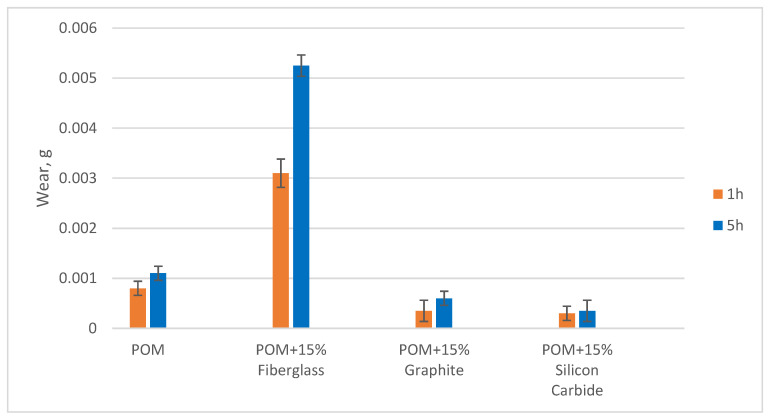
Volumetric wear of samples after hydroabrasion testing.

**Figure 11 polymers-16-00496-f011:**
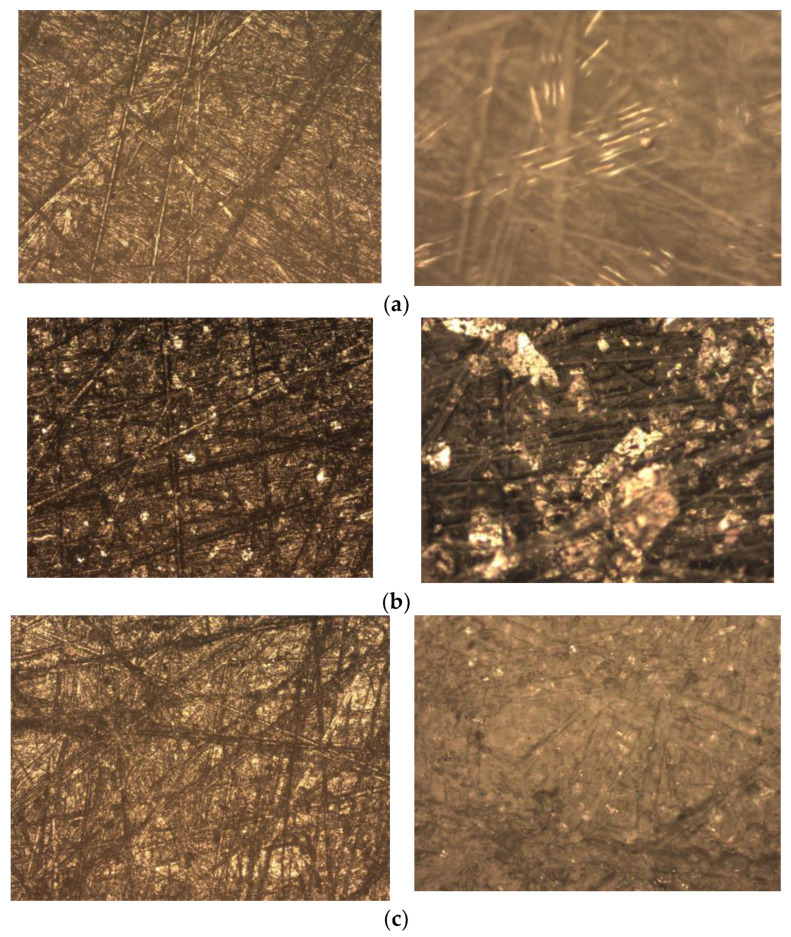
Microscopic results: (**a**)—POM filled fiber glass; (**b**)—POM filled graphite; (**c**)—POM filled silicon carbide.

**Figure 12 polymers-16-00496-f012:**
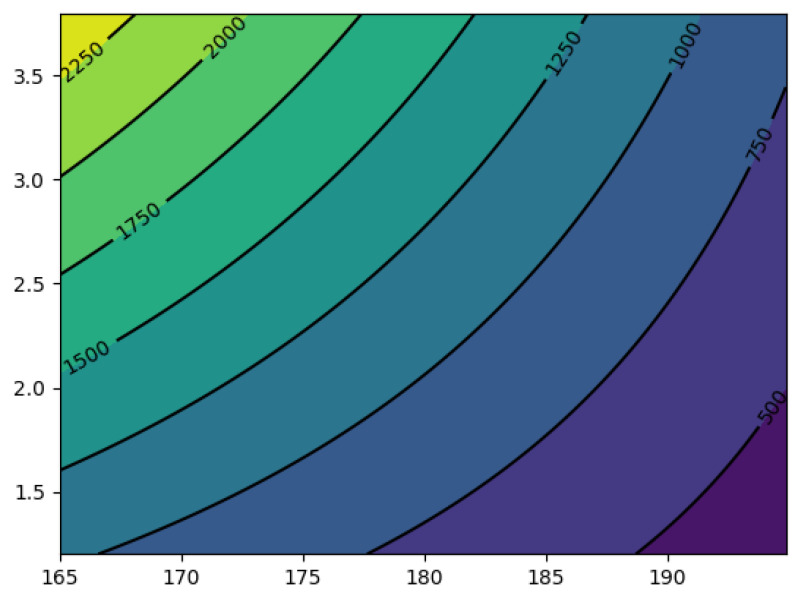
Response surface.

**Table 1 polymers-16-00496-t001:** Compositions of the mixtures tested.

Characteristic	Filler Type								
	Fibreglass			Silicon Carbide			Graphite		
Volume filling percentage, %	10	15	20	10	15	20	10	15	20
Particle size	l = 0.2 mm			d = 5–10 μm			d = 1.5 mm		

**Table 2 polymers-16-00496-t002:** Dimensions of test samples.

Size	Value, mm
Total length	73
Thickness working part	2
Length of the narrow part with parallel sides (working part)	35 ± 1
Large radius of curvature	25 ± 1
Small radius of curvature	14.0 ± 0.5
Head width	12 ± 1
Width of the narrow (working) part	4.0 ± 0.4

**Table 3 polymers-16-00496-t003:** Chemical concentrations during the test.

The Preparation	Consumption Rate	Volume of Water Used	The Amount of Preparation Needed	
			Standard Concentration	Increased Concentration
Herbicide “Lazurite”	0.5–1 kg/ha	400 mL	1 g	5 g
Fungicide “Rakurs”	0.2 L/ha	500 mL	0.5 mL	2.5 mL
Insecticide Borei Neo	0.1–0.2 L/ha	500 mL	0.5 mL	2.5 mL

**Table 4 polymers-16-00496-t004:** Weight measurement results for samples before and after the hydroabrasion test.

Sample Material	Sample Number	Weight of the Samples before the Test, g	Weight of the Samples after the Test, g	
			1 h	5 h
POM	1	0.2598	0.2591	0.2566
	2	0.2647	0.2638	0.2601
POM + 15% Fiberglass	1	0.3174	0.3145	0.3123
	2	0.3221	0.3188	0.3167
POM + 15% Graphite	1	0.3044	0.3042	0.3040
	2	0.3043	0.3040	0.3036
POM + 15% Silicon Carbide	1	0.3000	0.3001	0.2955
	2	0.3045	0.3041	0.3040

**Table 5 polymers-16-00496-t005:** Experimental data.

Number of Experience	Experimental Data						
	x_1_	x_2_	x_3_	y_1_	y_2_	y_3_	y_k_
1	165.00	3.80	10.00	2401.57	2477.56	2455.86	2445.00
2	195.00	1.20	10.00	410.30	428.00	403.21	431.84
3	165.00	1.20	20.00	1064.27	1131.56	1121.48	1105.77
4	195.00	3.80	20.00	925.88	880.12	891.57	899.19
5	165.00	1.20	10.00	952.73	983.21	962.90	966.28
6	195.00	3.80	10.00	695.77	718.95	718.95	711.22
7	165.00	3.80	20.00	2510.08	2336.34	2336.34	2394.25
8	195.00	1.20	20.00	445.66	459.76	459.76	455.06

where x_1_—test temperature, °C; x_2_—load, kg; x_3—_volume content of filler, %.

**Table 6 polymers-16-00496-t006:** Statistical evaluation of the planned experiment (0.95 confidence level).

Name	Value
Total dispersion	82,426.22
Maximum dispersion, Dk, max	68,365.15
Calculated Cochran criterion, *G_P_*	0.8294102
Tabular Cochran criterion, *G_T_*	0.5157
Homogeneity of dispersions	heterogeneous
$\mathrm{Adequacy}\;\mathrm{dispersion},\;D_{rel}^{2}$	1638.774
$\mathrm{Reproducibility}\;\mathrm{dispersion},\;D_{y}^{2}$	10,303.28
Fisher’s R-criterion Fp	0.1591
Fisher’s table criterion F_T_	$\mathrm{not}\;\mathrm{determined},\;(D_{rel}^{2}<D_{y}^{2}$)
Adequacy of the mathematic model	The model is adequate (Fp < 1)

## Data Availability

All data supporting the conclusions of this article are included in this article.

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
