# Peer review of "Testing of Polymer Composites for Manufacturing of Sprayer Nozzles"

_polymers, 2024, doi:10.3390/polym16040496_

Round 1
Reviewer 1 Report
Comments and Suggestions for Authors
Slavkina et al studied the durability and wear of the nozzle based on polymers, or filled ceramics etc. Then, this paper presents the wear properties of nozzles based on polymers nozzles and their chemical or abrasion resistance to the pesticides. The results are interesting and various experimental tests were performed and nicely supported by the discussion sections. Finally, the results are concluded in the conclusion. Overall, the paper is quite interesting, and within scope of the Polymers readership. However, the paper requires major revision before possible publication. Some points are –
[1] The title of the paper is not the true representative of this work and must be changed.
[2] In Figure 1, after (f) it should be (g) and not (j)? please check all such mistakes throughout the paper.
[3] The introduction is very confusing. In one way authors says ceramics are expensive and says cost is lower for polymer based nozzles but then they talk about the disadvantages about polymers based nozzles? This is very confusing. Please rewrite this information again.
[4] The novelty of the paper must be further described in last paragraph of introduction. Authors should make clear agenda about the work, why this work is important, what challenges in the field can be solved by this work? What are its advancement of this work with existing literature?
[5] Figure 4 is of no use. What do authors want to tell from such images? What type of corrosive environment authors want to tell? Is it acidic environment? If authors want to study corrosive degradation, they should in fact use FTIR or other spectroscopic techniques for such studies?
[6] In Figure 6, could it be supported with optical image of the device?
[7] All the equations used in the work should be supported with references wherever applicable.
[8] The presentation of the figures is rather very poor. The figures should be supplied with better resolution? Finally, all Figure captions need to be re-written?
[9] The conclusion need to be re-written. It should be focus on the experimental outcomes; key take away from the experiments. Finally, current prospects and future prospects should be reported.
Good Luck with the revisions!
Comments on the Quality of English LanguageModerate editing of English language required.
Author Response
Good afternoon. We send You the manuscript edited according to Your remarks.
[1] The title of the paper has been changed to a more applied title based on the recommendations of reviewer 2.
[2] The letter designations in Figure 1 and all subsequent figures have been checked and corrected.
[3] Agreed with the comment there were missing points where information was not presented consistently or was repetitive. This comment has been modified by structuring the previously written introduction.
[4] This comment has been corrected. Information has been added at the end of the introduction.
[5] Figure 4 has been deleted.
[6] Figure 6 has been supplemented.
[7] All equations used are generally known. All information about them can be found on the Web.
[8] The quality of the drawings has been improved where possible.
[9] The conclusion has been finalised. It describes the main results of the work carried out.
Thank you for your notes, hopefully we were able to correct the manuscript in its entirety.
With best wishes,
Alexey Kutyrev
Reviewer 2 Report
Comments and Suggestions for Authors
The authors propose to use polymeric nozzles for spraying of chemicals in plant protection. They investigated possible polymeric compositions and derived conclusions about the optimum composition for their purposes. The paper seems to be interesting to me, with clear application area. However, there are several weakness that have to be addresses. Please find my comments below.
1) On Figure 2, I cannot see the data for ceramics.
2) Equation (1): Why is n in the denominator and not (n-1)? How do you calculate confidence interval? At which confidence level? Please provide more details? Do you follow "the guide to uncertainty in measurements"?
3) Figure 10: There are some jumps in the weight of the samples in both directions. How can you explain them? Are they statistically significant or these are just measurement errors?
4) The title of your paper seems to be somewhat misleading to me. I expected a different contnet. I suggest to modify the title in such a way that it will more reflect the applied character of your research.
Comments on the Quality of English Language
It should be edited for clarity.
Author Response
Good afternoon. We send You the manuscript edited according to Your remarks.
[1] Ceramics data was submitted, column signatures moved out. The deficiency has been corrected.
[2] There is full agreement with the observation. The previous equation was a calculation of standard deviation only. The observation has been corrected and the confidence level is given (95%).
[3] These mass jumps may be due to errors in the experiment. On average, the mass variation of the samples varies within 0.1-0.7% and does not change significantly, so these errors do not affect the final data.
[4] The title of the article has been changed as per the comment.
Thank you for your notes, hopefully we were able to correct the manuscript in its entirety.
With best wishes,
Alexey Kutyrev
Round 2
Reviewer 1 Report
Comments and Suggestions for Authors
Please replot all the graphs in high resolution (>300 dpi) before publication.
Author Response
Good afternoon, the quality of the Figures has been improved.
Regards,
Alexey Kutyrev
Reviewer 2 Report
Comments and Suggestions for Authors
The authors have replied to my comments. They are OK exept reply #3:
"These mass jumps may be due to errors in the experiment. On average, the mass variation of the samples varies within 0.1-0.7% and does not change significantly, so these errors do not affect the final data."
--> This is exactly the point. You did´t draw the error bars on the corresponding Figures. Therefore, it is not possible to see if the differences are within the uncertainity of measurements or not. For me, it seems like the mass jumps are significantly higher than 0.1 - 0.7% said. State the uncertainiy of mass measurements in text explicitly and explain how it was obtained. Please improve your presentation. If the jumps are not witin your claimed uncertainty please exlain the reason.
Comments on the Quality of English LanguageI
Author Response
Good afternoon, thank you for your comment. We have checked the data and there was an error when processing the results. You are correct, the weight change was up to 4.5%. All changes have been made and a confidence interval has been added to the graph.
Regards,
Alexey Kutyrev